# Identification and Expression Analysis of the *SKP1-Like* Gene Family under Phytohormone and Abiotic Stresses in Apple (*Malus domestica*)

**DOI:** 10.3390/ijms242216414

**Published:** 2023-11-16

**Authors:** Miao Shao, Ping Wang, Huimin Gou, Zonghuan Ma, Baihong Chen, Juan Mao

**Affiliations:** College of Horticulture, Gansu Agricultural University, Lanzhou 730070, China

**Keywords:** apple, MdSKP1-Like, abiotic stress, phytohormone, identification, expression analysis

## Abstract

Ubiquitination participates in plant hormone signaling and stress response to adversity. SKP1-Like, a core component of the SCF (Skp1-Cullin-F-box) complex, is the final step in catalyzing the ubiquitin-mediated protein degradation pathway. However, the *SKP1-Like* gene family has not been well characterized in response to apple abiotic stresses and hormonal treatments. This study revealed that 17 *MdSKP1-Like* gene family members with the conserved domain of SKP1 were identified in apples and were unevenly distributed on eight chromosomes. The *MdSKP1-Like* genes located on chromosomes 1, 10, and 15 were highly homologous. The *MdSKP1-like* genes were divided into three subfamilies according to the evolutionary affinities of monocotyledons and dicotyledons. *MdSKP1-like* members of the same group or subfamily show some similarity in gene structure and conserved motifs. The predicted results of protein interactions showed that members of the MdSKP1-like family have strong interactions with members of the F-Box family of proteins. A selection pressure analysis showed that *MdSKP1-Like* genes were in purifying selection. A chip data analysis showed that *MdSKP1-like14* and *MdSKP1-like15* were higher in flowers, whereas *MdSKP1-like3* was higher in fruits. The upstream *cis*-elements of *MdSKP1-Like* genes contained a variety of elements related to light regulation, drought, low temperature, and many hormone response elements, etc. Meanwhile, qRT-PCR also confirmed that the *MdSKP1-Like* gene is indeed involved in the response of the apple to hormonal and abiotic stress treatments. This research provides evidence for regulating *MdSKP1-Like* gene expression in response to hormonal and abiotic stresses to improve apple stress resistance.

## 1. Introduction

Ubiquitin (Ub) is a small protein consisting of 76 amino acids, that is highly conserved in eukaryotes. Ubiquitination plays a key role in protein function, regulation, and degradation. Plants can thus rely on ubiquitination in the environment to regulate hormone signaling and stress responses [1]. The process of ubiquitination in plants is mainly regulated by a variety of ubiquitinases, such as ubiquitin-activating enzymes, binding enzymes, ligases, and dissociative enzymes (E1, E2, E3, and DUBS) [2,3]. E3 ligases act as a specific substrate selector in the ubiquitin protease cascade reaction. Structurally, E3 ligases can be divided into single-subunit E3 ligases and multi-subunit E3 ligases [4]. SCF (SKP1/Cullin1/F-box) [5] complex is a class of multisubunit RING E3 ligases consisting of the SKP1 protein, the Cullin1 protein, the RBX1 protein, and a variable F-box protein [6]. Previous work found that FBX proteins can interact with ASK proteins in *Arabidopsis* [7]. The s-phase kinase-related protein 1 (SKP1), a major component of the SCF complex, plays an important role in ubiquitin-mediated plant-protein degradation [4].

ASK1 was involved in protein degradation during flowering in *Arabidopsis* [8]. ASK1 can also act with other members of the SCF complex to control plant light signals [9] and was related to the plant’s viral defense [10]. Currently, 21,31,19, and 15 *SKP1-Like* genes were firmly established in *Arabidopsis* [11], rice [12], tomato [13], and chickpea [14] species, respectively. *SKP1-Like* genes have also been identified in several species, such as wheat *SKP1* genes, soybean *SKP1* genes, pepper *SKP1* gene, citrus *SKP1* gene, and peony *SKP1* gene [15,16,17,18,19].

The SCF complex formed by SKP1 was in response to many hormone regulatory pathways, and TIR1 and ASK1 interacted to form the SCF^TIR1^ complex, which can regulate Aux/IAA proteins in the hormone-signaling pathway of auxin [20], while the SCF^SLEEPY1^ complex formed by SLEEPY1 and ASK1 was involved in the regulation of the hormone-signaling pathway of gibberellin [21]. In *Arabidopsis*, COI1 can act as a jasmonate receptor [22] and some ASK can bind to it to form the SCF^COI1^ complex, which is involved in the jasmonate hormone-signaling pathway [23]. SKPs could be related to the hormone signaling pathway of ABA and, thus, respond to abiotic stresses [24,25].

Extensive studies have proven that *SKP1-Like* genes participated in abiotic stresses in plants. For example, *SSK* genes in wild tomato plants responded to both heat and salt stress [13]. In soybean (*Glycine max*), the *GmSK1* gene overexpression resulted in increased plant tolerance to high salt and drought stresses [18]. *GsSKP21*, a *SKP1-Like* family gene, has a conserved SKP structural domain. *GsSKP21* gene overexpression enhanced soybeans’ (*Glycine soja*) tolerance to alkaline stress and also reduced plant sensitivity to ABA [26]. In peonies, the *PSK1* gene overexpression also enhanced the drought tolerance of plants and promoted flower formation and early flowering [17]. In chickpeas, *CaSKP* genes are also involved in drought, salt, and oxidative stresses in plants [14]. These suggest that *SKP1-Like* genes may play a crucial role in apple response to abiotic stresses and phytohormone signaling pathways. Therefore, genome-wide exploration of the *MdSKP1-Like* gene family in response to environmental stresses may provide insights into the precise regulation of tolerance to various environmental stresses and responses to exogenous hormones in apple.

Apples are a major source of fresh food and vitamin supplementation for people daily. However, apple growth is threatened by drought, salinity, and osmotic stress, etc. In this study, the apple *SKP1-Like* gene family was analyzed and identified using bioinformatics methods. The distribution, gene structure, gene homology, conservation patterns, selection pressure, codon preference analysis, predicted-protein interaction pathways, cis-acting elements, and evolutionary relationships of *SKP1-Like* genes in apple chromosomes were analyzed at the genomic level. The expression levels of *SKP1-Like* genes in different apple tissues were analyzed using relevant databases. Many cis-regulatory elements associated with phytohormones and abiotic stresses were identified in the promoter of *MdSKP1-Like* genes. Therefore, we performed the qrt-PCR to analyze the *MdSKP1-Like* gene family expression in different hormone treatments and abiotic stresses.

## 2. Results

### 2.1. Phylogenetic Tree Analysis of SKP1-Like Gene

The phylogenetic analysis of SKP1-Like proteins of six species were carried out using MEGA7. The results showed (Figure 1) that all SKP1-Like family members could be divided into three subfamilies. Among them, subfamily Ⅰ contained MdSKP1-Like2, MdSKP1-Like4 and MdSKP1-Like6; subfamily II contained MdSKP1-Like5, MdSKP1-Like8, MdSKP1-Like15, MdSKP1-Like16 and MdSKP1-Like17; and the rest of the MdSKP1-Like family members belonged to subfamily Ⅲ. Analysis of the subfamilies showed that subfamily I and subfamily III contained each of the six species studied, while subfamily II did not contain OsSKP1-Like members. Evolutionary relationships of subfamilies I and III indicate that members of the apple SKP1-Like family are more closely related to members of the strawberry SKP1-Like family members. Evolutionary relationships of subclade II indicate that apple SKP1-Like family members are more closely related to grape and strawberry SKP1-Like family members compared to other species. The above results suggest that plant SKP1-Likes have different characteristics in evolution, with apple SKP1-Like family members being more closely related to those of the dicotyledonous plants strawberry, grape, and *Arabidopsis*, and more distantly related to monocotyledonous plant rice and *Ananas comosus* SKP1-Like members in evolution.

### 2.2. Analysis of Physicochemical Properties and Chromosomal Localization

A total of 17 non-redundant *MdSKP1-Like* genes were identified from the whole apple genome, and their deduced amino acid sequences all contained typical Skp1 structural domains. According to their positions on chromosomes, the *MdSKP1-Like* genes were named *MdSKP1-Like1-17* (Appendix A); the 17 *MdSKP1-Like* genes were unevenly distributed on eight chromosomes (Figure 2A). A total of four gene clusters were found, such as chromosomes 1, 8, 10 and 15. Four genes distributed on chromosome 10 were *MdSKP1-Like11*, *MdSKP1-Like12*, *MdSKP1-Like13*, and *MdSKP1-Like14*. There are four genes distributed on chromosome 1 *MdSKP1-Like1*, *MdSKP1-Like2*, *MdSKP1-Like3*, and *MdSKP1-Like4. MdSKP1-Like16* and *MdSKP1-Like17* were distributed on chromosome 15, while *MdSKP1-Like5, MdSKP1-Like6, MdSKP1-Like7,* and *MdSKP1-Like15* were distributed on chromosomes 2, 4, 5 and 12, in that order. Also based on gene density, it was seen that *MdSKP1-Like4*, *MdSKP1-Like8,* and *MdSKP1-Like16* were located in high-density regions. The average amino acid size, molecular weight/KD, isoelectric point, and instability index of proteins from members of subfamily Ⅰ are higher than those of members of the other two subfamilies (Figure 2B). The longest corresponding protein consisted of 356 amino acid residues (MdSKP1-Like2) and the shortest consisted of 108 amino acid residues (MdSKP1-Like1), the molecular weight (D) is 12147 (MdSKP1-Like1)~41130 (MdSKP1-Like2), and the isoelectric point (pI) is 4.27 (MdSKP1-Like16)~5.21 (MdSKP1-Like2) (Appendix A). The instability indices of these proteins ranged 30.86 (MdSKP1-Like12)~56.98 (MdSKP1-Like4), and we also found that these proteins are hydrophilic. The grand average of hydropathicity ranged −0.019 (MdSKP1-Like3)~−0.782 (MdSKP1-Like2), and the aliphatic index ranged 70.74 (MdSKP1-Like10)~110.14 (MdSKP1-Like3) (Appendix A).

### 2.3. The Subcellular Location Prediction and Secondary Structure Analysis

Subcellular localization of *MdSKP1-Like* genes analyses showed (Figure 3A) that the *MdSKP1-Like* genes were mainly located in the nuclear and cytoplasmic areas. Among them, *MdSKP1-Like2*, *MdSKP1-Like4,* and *MdSKP1-Like6* were predicted to be predominantly distributed in the nucleus not in the lysosome. *MdSKP-Like5*, *MdSKP1-Like8*, *MdSKP1-Like10*, *MdSKP1-Like14*, *MdSKP1-Like15,* and *MdSKP1-Like17* were predicted to have a high distribution in both the nucleus and cytoplasm, with most of the remaining members predicted to be predominantly distributed in the cytoplasm.

Apple SKP1-Like protein secondary structure (Figure 3B) showed that the secondary structure mainly consisted of alpha helix, extended strand, beta turn, and random coil. Alpha helix was mainly distributed between approximately 44.66 and 70.37%, the extended strand was mainly distributed between approximately 1.48 and 8.86%, the beta turn was mainly distributed between approximately 2.22 and 6.48%, and the random coil was mainly distributed between approximately 18.52 and 43.26%.

### 2.4. The MdSKP1-Like Gene Synteny Analysis

The *MdSKP1-Like* gene family synteny analysis results showed that three pairs of collinear relationships were found, namely *MdSKP1-Like3*/*MdSKP1-Like6*, *MdSKP1-Like4*/*MdSKP1-Like10*, and *MdSKP1-Like2*/*MdSKP1-Like16* (Figure 4A). The results show that *MdSKP1-Like* genes may undergo amplification of family members through gene duplication during the evolutionary process. To further predict the phylogenetic components of the *MdSKP1-Like* family, we constructed interspecies collinear-relationship gene analysis maps for apple and *Arabidopsis*, rice, strawberry, and grape (Figure 4B). Among them, Apple had 14, 12, 10, and 1 homologous pairs of genes with strawberry, *Arabidopsis*, grape, and rice, respectively. Overall, *MdSKP1-Like* was more distantly related to *OsSKP1-Like* and more closely related to strawberry. Since they were more closely related genetically, it was hypothesized that *MdSKP1-Like* and *FvSKP1-Like* share some similarities in functional expression.

### 2.5. Gene Structure, Motif Composition and Structural Domain Analysis of MdSKP1-Like

Based on evolutionary tree relationships, the apple *SKP1-Like* gene family was divided into three groups (Figure 5). Group Ia consists of *MdSKP1-Like5*, *MdSKP1-Like17*, *MdSKP1-Like7, MdSKP1-Like12, MdSKP1-Like16, MdSKP1-Like8,* and *MdSKP1-Like15*, all of which contained conserved motif 1/2/4/6, and conserved structural domains skp1 and BTB-POZ-SKP1. Group IIb consisted of *MdSKP1-Like4, MdSKP1-Like6, MdSKP1-Like14, MdSKP1-Like9,* and *MdSKP1-Like10*, all of which contained motif 4 and the conserved structural domain skp1. The members of group IIIc consisted of *MdSKP1-Like1, MdSKP1-Like2, MdSKP1-Like3, MdSKP1-Like11,* and *MdSKP1-Like13.* Among them, *MdSKP1-Like11* and *MdSKP1-Like13* contain conserved motif 1/2/3/4/6, which were consistent with group Ia members. All members of the *MdSKP1-Like* family contain the conserved structural domain Skp1. As shown in Figure 5, the *MdSKP1-Like* gene’s introns and exon numbers were different in different subfamilies, with the exon numbers ranging from 1 to 14. *MdSKP1-Like4* and *MdSKP1-Like2* have the largest number of exons, 14 each, and *MdSKP1-Like2* has the longest gene, about 64 kb, and *MdSKP1-Like1* has the gene that was the shortest. The number and distribution of introns and exons within the same group are also different, and only a few of them have relatively high conservation. The structures of *MdSKP1-Like7* and *MdSKP1-Like12* in group Ia were similar; the structures of *MdSKP1-Like11* and *MdSKP1-Like13* in group IIIc were similar. It was hypothesized that *MdSKP1-Like* gene members with similar gene structures in each group perform the same functions among themselves.

### 2.6. Evolutionary Selection Pressure and Codon Usage Bias Analysis

We analyzed the Ka/Ks values of apple and *Arabidopsis* collinear relationship genes. The results showed that three gene pairs of *MdSKP1-Like* (Figure 6A) and six gene pairs of *AtSKP1-Like* had Ka/Ks values less than one (Figure 6B). This indicated that the *SKP1-Like* family members in apple and *Arabidopsis* were mainly under purifying selection. Among the analyzed codon usage deviations, the largest Nc value was 58.59 for *MdSKP1-Like1* and the smallest Nc value was 39.53 for *MdSKP1-Like12*. The CAI values of *MdSKP1-Like* family members ranged from 0.15 to 0.37, and the frequencies of Fop ranged from 0.31 to 0.54 (Figure 6D). Compared to the values of A3s and T3s, we found larger values for GC3s and GCs for *MdSKP1-Like* family members (most values were around 0.5). The GC3s in the *MdSKP1-Like* family were positively related to CBI and Fop, but the correlation between T3S, A3S, CBI, and Fop in the *MdSKP1-Like* family was negative. (Figure 6E). RSCU is an assessment of the preference for the use of synonymous codons. Statistics revealed that GUC-encoded Val was used most frequently and with the greatest preference. CUA-encoded Leu was used least frequently and with the least preference (Figure. 6C). No preference was found for CAU and CAC-encoded His, AUG-encoded Met, CCA-encoded Pro, and UGG-encoded Trp, respectively.

### 2.7. Analysis of Protein Interactions of MdSKP1-Like Family Members

The protein interactions of MdSKP1-Like family members were predicted using the STRING website with *Arabidopsis* as the model plant. The results showed that the MdSKP1-Like8 homologous protein ASK2 (AT5G42190) (Appendix A) interacted with AT1G76920, AT4G05460, galactose oxidase (ZTL), EIN3-binding F box protein 1 (EBF1), cullin1 (CUL1), regulator of cullins-1 (RBX1), corinsensitive 1 (COI1), and F-Box protein 7 (FBP7), etc (Figure 7A). Meanwhile, MdSKP1-Like11/12/15/16 were not involved in protein interactions, while the other members were involved in protein interactions with RBX1, FBP7, and other proteins interacted closely, indicating that the MdSKP1-Like family members exerted diverse functions in performing biological functions (Figure 7B), not only by having interactions among family members but also by interacting with many different proteins (especially the F-Box) at the same time.

### 2.8. Cis-Acting Elements Analysis of Apple MdSKP1-Like Gene Family

Using *MdSKP1-Like* gene upstream 2kb promoter to predict the cis-acting element. The results showed that the *MdSKP1-Like* gene promoter was related to hormones and stress (Figure 8). The hormone response element contains methyl jasmonate (CGTCA-motif, TGACG-motif), salicylic acid (TCA-element, as-1), auxin cis-elements (TGA-element), gibberellin (TATC-box), and abscisic acid (ABRE). We also identified many cis-regulatory elements associated with stress conditions, for example, regulatory elements involved in biotic and abiotic stresses in plants (MYB), MYB binding sites MBS and MYC. Other stress-specific cis-elements include drought-stress, salt-stress-sensitive reaction element (DRE), low-temperature reaction element (LTR), temperature-responsive stress reaction element (STRE), and anaerobic response element (ARE). Among them, the action elements related to salt stress, which were DRE and ABRE, the action element related to low temperature response (LTR), and the hormone element related to plant flowering (TATC-box). Notably, *MdSKP1-Like8* was enriched for these four types of response elements simultaneously.

### 2.9. Analysis of MdSKP1-Like Gene Expression Based on GEO Database

Expression analyses of 17 *MdSKP1-Like* genes in seedling and different plant organs were performed by searching the GEO database. As shown, *MdSKP1-Like8, MdSKP1-Like5,* and *MdSKP1-Like17* were expressed at high levels in all plant organs and seedling, and the remaining 14 genes were expressed at certain levels in different plant organs (Figure 9). Among them, *MdSKP1-Like14* and *MdSKP1-Like15* had a higher expression in flowers compared with other plant organs. Notably, other members of the *MdSKP1-Like* gene family, except *MdSKP1-Like13*, were expressed at higher levels in flowers, which, combined with the gibberellin regulatory element found in the upstream 2 KB of the promoter, predicts that *MdSKP1-Like* family members played a regulatory role in the growth and development of the flower.

### 2.10. Expression of MdSKP1-Like Gene in Apple Treated with Exogenous Hormones and Abiotic Stress Treatment

The above results indicated that the *MdSKP1-Like* gene may involve multiple stress-related cis-acting elements. After 24 h of treatment at 4 °C, all *MdSKP1-Like* members’ expression was below the control (Figure 10). After 24 h of PEG treatment, the expression of *MdSKP1-Like2*, *MdSKP1-Like10*, and *MdSKP1-Like13* was higher than the control, and other members’ expression was lower than the control. The sensitivity of *MdSKP1-Like* genes to 24 h of NaCl treatment was higher than the control. Among them, the expression level of *MdSKP1-Like9* was the highest. We also detected many elements related to hormone regulation at 2 kb upstream, so we could treat apple plants with different hormones, and the *MdSKP1-Like* gene expression differed significantly under different hormone treatments (Figure 10). Under MeJA treatment, the relative expression of *MdSKP1-Like9* appeared to be downregulated and the rest of the genes were upregulated to some extent. After ABA treatment, the relative expression of *MdSKP1-Like9* and *MdSKP1-Like1* appeared to be downregulated and the rest of the genes were upregulated. Notably, the expression of *MdSKP1-Like3* was highest under all three hormone treatments, respectively.

## 3. Discussion

### 3.1. Evolutionary Properties of the MdSKP1-Like Family

The *SKP1-Like* gene family was identified in many organisms, for example 21, 21, 31, and 19 members were identified from *Cryptomeria hidrophila* [21], *Arabidopsis thaliana* [12], rice [16], and tomato [13], respectively. The identification and systematic analysis of the *SKP1-Like* gene in apples is not clear. In this study, 17 *SKP1-Like* family members were identified from the apple. The physical- and chemical-property analysis shows that each member contains the conserved structural domain SKP1 in apples (Figure 5). Subcellular localization predicted that most *MdSKP1-Like* genes were located in the nucleus and cytoplasm, which corresponded to the *SKP1-Like* gene of *Arabidopsis* [27]; therefore, it was predicted that the *MdSKP1-Like* gene family was important for controlling the metabolism and genetic material of apple cells. The chromosomal localization map indicated that most *MdSKP1-Like* genes were located on the 10th chromosome (Figure 2A), in agreement with the results of previous studies on tomato [13]. All these results indicate that *SKP1-like* genes are conserved across the evolution of different species. This experiment also revealed that *MdSKP1-Like* genes located on chromosomes 1, 4, 10, and 15 have high homology (Figure 4A).

Previous experiments have shown that both the *ASK1* gene in *Arabidopsis* and the *OSK1* gene in rice originated from a single ancestor and that gene duplication provided the raw material for an evolutionary process that allowed these genes to perform similar functions [12,28]. Gene duplication is the main driver of gene family expansion, allowing the gene family to acquire new functions and evolve. Gene replication includes fragment replication, tandem replication and genome replication, with fragment replication being more beneficial for maintaining gene function [29]. The results showed that members of the *MdSKP1-Like* family undergo family-member amplification through fragmental replication during the evolutionary process. Previous studies divided the *CaSKP1-Like* family into three subfamilies based on genes with a single intron, lack of introns, and genes with a large number of introns at different locations [14]. In contrast, the *MdSKP1-Like* gene family was constructed from the monocotyledons and dicotyledons perspective in this study and was divided into three subfamilies, of which 3, 5, and 9 *MdSKP1-like* genes were sequentially distributed in subfamilies I, II, and III (Figure 1). The apple *SKP1-Like* family members were more closely related to dicotyledons (grape, strawberries) and more distantly related to monocotyledons (rice). The interspecies collinear relationship gene analysis also verified that *MdSKP1-Like* had more collinear relationships gene pairs with *SKP1-Like* family members of strawberry, and presumably the collinear members have similar biological functions, which also indicates to some extent that the apple was more closely related to *SKP1-Like* family members of strawberry (Figure 3B). These results may be because strawberry and apple belong to the same Rosaceae family.

The gene-selection pressure and codon bias analysis also contribute to the understanding of evolutionary relationships [30,31]. In this study, we calculated the intraspecific collinear relationship gene Ka/Ks ratios for *Arabidopsis* and apple, respectively, and most of them were less than one. The above results suggest that *SKP1-Like* family members in *Arabidopsis* and apple experience purifying selection effects. In addition, the codon bias analysis of *MdSKP1-Like* showed that GC3s in the *MdSKP1-Like* family were positively related to CBI and Fop, but the correlation between T3S, A3S, CBI, and Fop in the *MdSKP1-Like* family was negative (Figure 6). This suggests that the base type at position three of the synonymous codon of the *MdSKP1-Like* gene affects the magnitude of the degree of codon usage preference.

### 3.2. MdSKP1 May Interact with F-Box and May Respond to Hormone-Signaling Pathway

It was found that the protein interaction between SKP1 and F-Box was confirmed in an increasing number of species [27,32]. For example, ASK was able to interact with AtTLP9 and RCAR3 in *Arabidopsis* [33], and tomato SSK interacts with COI1 and TIR1 [13]. Furthermore, SKP1 has been shown to interact with F-Box in a non-SCF complex [34,35]. Results of protein interaction prediction in this study showed that most of the *MdSKP1-Like* family members may interact with FBP7, F-Box, and RBX1 proteins (Figure 7), but the specific mechanism of action needs to be further explored. The promoter analysis of *MdSKP1-Like* family members showed that *MdSKP1-Like* contains multiple hormone response elements. Among them, abscisic acid (ABA), methyl jasmonate (MeJA), and salicylic acid (SA) response elements were more abundant. It was suggested that ABA, MeJA, and SA may be important in the hormone response of the *MdSKP1-Like* family [23,36]. In SA or ABA treatments, nearly 50% of tomato *SSK* gene levels were upregulated, while under MeJA treatments most *SSK* genes were downregulated [13]. *GMSK1* expression was upregulated in soybean under SA, ABA, and MeJA treatment after 48 h [18]. Following the same, we conducted different hormone treatments on apples, and the results showed that most *MdSKP1-Like* genes were upregulated after SA or ABA treatment, but *MdSKP1-Like* genes were also upregulated but to a lesser extent after MeJA treatment.

The Phylogenetic relationship between *MdSKP1-like8* and ASK2 (AT5G42190) (Appendix A) was the closest [11]. Based on the fact that the *Arabidopsis ASK* gene can regulate plant flowering [8], gibberellin was found to promote flowering in *Arabidopsis* by degrading DELLA [37], and the analysis of the upstream cis-acting elements of *MdSKP-like8* revealed more gibberellin elements (TATC-box); it was speculated that *MdSKP-like8* may regulate plant flowering. In combination with the previous study that found that ASK1 can regulate plant flowering and plant growth [38], it was hypothesized that these genes are important in regulating apple’s growth and development. It was also surprising to find high relative expression of most group Ia members in plant floral tissues (classified according to conserved motifs) (Figure 5), and more gibberellin-acting elements found in the node and presumably group Ia members may participate in the regulation of flower growth and development (Figure 9). Different tissue’s expression analyses showed that 17 *MdSKP1-Like* gene expressions were different in different plant tissues, and this difference in transcriptional expression could be attributed to the differentiation of cis-elements in promoters [39].

### 3.3. Structural Analysis of MdSKP1-Like Gene and Speculation on Its Function in Abiotic Stress Treatment

The distribution position of exons on the same group of amino acid sequences was highly variable, and there was no obvious pattern in sequence length. In this paper, 17 *MdSKP1-Like* genes were predicted to have nine conserved motifs (Figure 5), and similar results were also obtained in tomato *SKP* genes [13]. In the group Ia, the conserved motifs of all family members were consistent, indicating a high degree of conservation among the genes in the *MdSKP1-Like* group Ia. The results of the motif analysis showed that *MdSKP1-Like2* and *MdSKP1-Like4* contained two special motifs (motif 8, motif 9) at the C-terminus. Combined with a similar action element of 2000 bp upstream of *MdSKP-Like2/4*; these results imply that *MdSKP1-Like2* and *MdSKP1-Like4* may have specific functions that distinguish them from other members of the family. The number, length, and distribution of exons were not regular at the protein code level, but the degree of conservation among members was high at the protein level. Different members of the *MdSKP1-Like* family adopted different transcriptional and translational processes to finally form mature proteins with similar functions indicating that the degree of conservation at the genes of this family was not consistent with that at the protein level.

*MdSKP1-Like* cis-acting elements analyses indicated that these elements such as low-temperature and drought are present in large numbers, indicating their potential function in plant response to environmental stresses. Previously, many studies have also demonstrated the involvement of the SCF complex in the ABA pathway [40]. The receptor proteins of the SCF complex, DWA1 and DWA2, both of which exhibit high sensitivity to ABA and the ability of DWA1 and DWA2 to bind ABA insensitive 5 (ABI5) in vivo to degrade ABI5 [41]. Based on the discovery of a large number of ABRE cis-elements, it was hypothesized that the *MdSKP1-Like* gene may be involved in abiotic stress responses mediated by the ABA pathway. Extensive studies have demonstrated that *SKP1* genes can enhance salt tolerance in pepper [19], and peony [17]. The relative expression of *GmSKP1* in soybean reached its highest level at 24 h of NaCl treatment [18]. Likewise, under NaCl treatment, the relative expression of *CaSKP1*, *CaSKP2*, *CaSKP5*, *CaSKP7*, *CaSKP8*, *CaSKP10,* and *CaSKP13* was upregulated in chickpea root systems [14]. In apple leaves treated with NaCl for 24 h, we found that *MdSKP1-Like1*, *MdSKP1-Like5*, and *MdSKP1-Like9* were significantly upregulated. Under drought stress, all genes were downregulated, except for *SSK5*, *SSK7,* and *SSK18* in tomato leaves. In chickpea roots and stems, we also found that most *CaSKP* genes were downregulated under drought stress [14]. We obtained similar results in apples, where most gene expression was downregulated after drought stress besides *MdSKP1-Like2*, *MdSKP1-Like10*, and *MdSKP1-Like13.* Most tomato *SKP1-Like* gene expression was downregulated under 4 °C temperature treatment [13]. Under 4 °C temperature treatment, all *MdSKP1-Like* gene expression was downregulated. These findings reveal the complexity of *MdSKP1-like* gene regulation and lay the foundation for further studies on the function of apple *SKP1-like* genes.

## 4. Materials and Methods

### 4.1. Plant Materials and Stress Treatment

The seedlings of the ‘Gala’ apple (*Malus* × *domestica*) were provided by the Fruit Seedling Research Laboratory of Gansu Agricultural University. After 30 d of succession culture, the seedlings were selected from healthy and uncontaminated cultures, and were successively treated with 10% PEG, at 4 °C low temperature, and with 150 mM NaCl. At the same time, plants with normal growth were used as the control. We treated the plants with each of the three hormones at 0.2 mM ABA, 5 mM SA, and 0.1 mM MeJA, and the same plants were treated with an equal distilled water as the control [18,42]. Three biological replicates were set up for each seedling’s treatment. Seedlings of uniform size were selected and the 4th~7th leaves from the base upward were quickly ground in liquid nitrogen to extract total RNA from the leaves using the CTAB method.

### 4.2. Identification and Characterization of Apple SKP1-Like Genes

The *SKP1-Like* gene family accession numbers of *Oryza sativa* L. and *Arabidopsis* were obtained from the relevant literature [11,12], and the corresponding full length CDS and full length genomic sequences were obtained in the tair databases (https://www.Arabidopsis.org, accessed on 16 September 2022) and rice genome databases (http://www.ricechip.org/, accessed on 20 September 2022). The obtained CDS sequences were compared using BLAST in the apple genome database (https://www.rosaceae.org/species/malus/all, accessed on 24 September 2022) [43], with the screening condition E ≤ 10^−10^. The screened genes were used in the apple genome database to search for the accession numbers, CDS sequences, protein sequences, and chromosomal location of predicted *MdSKP1-Like* gene family obtained using the homologous search. All predicted proteins were examined for structural integrity using the SMART (http://smart.embl-heidelberg.de/, accessed on 2 October 2022) online website [44], and sequences without the characteristic structural domains were removed to obtain the *MdSKP1-Like* genes. Protein’s physical and chemical properties were predicted using the online software ExPASy (https://web.expasy.org/protparam/, accessed on 4 October 2022) [45]. Using the online software CELLOV2.5 (http://cello.life.nctu.edu.tw/, accessed on 8 October 2022) [46], we predicted the subcellular localization of *MdSKP1-Like* genes.

### 4.3. Construction of a Phylogenetic Tree, Chromosome Localization, and Synteny Analysis

The ClustalX program was used to compare amino acids of *SKP1-Like* gene families of dicotyledonous plants such as apple, strawberry, grape, and *Arabidopsis,* and monocotyledonous plants such as rice and bromeliad to classify the *MdSKP1-Like* gene family according to monocotyledonous species. The evolutionary tree analysis software MEGA7.0 and the neighbor-joining method (NJ) were used to construct an evolutionary tree [47]. And phylogenetic tree embellishment was performed with Itol (https://itol.embl.de, accessed on 11 October 2022). *MdSKP1-Like* gene family location mapping was generated using TBtools (https://github.com/CJ-Chen/TBtools). Construction of synteny analysis maps between apples and other species using TBtools (version 1.108)

### 4.4. Construction of Gene Structure, Motif Sequence Analysis and Structural Domains

*MdSKP1-Like* gene’s exon and intron structures were obtained in the software TBtools (version 1.108) [48]. The MEME (http://meme.nbcr.ne-t/meme/cgi-bin/meme.cgi, accessed on 15 October 2022) set as any number of repetitions and the number of predicted motifs was 9 was used to analyze the protein conserved motifs of the MdSKP1-Like gene family. The data of the conserved structural domains of the apple SKP1-Like gene family were also obtained from the NCBI website (https://www.ncbi.nlm.nih.gov, accessed on 15 October 2022) [49] and mapped in the software TBtools.

### 4.5. Analysis of Selective Pressure and Codon Usage Index

The non-synonymous substitution rate and synonymous substitution rate of the *MdSKP1-Like* gene with co-linear relationships were calculated using the NG method of TBtools and mapped using Origin 2021. The CDS sequences of *MdSKP1-Like* were obtained and the codon usage characteristics of these genes were analyzed by CodonW software [50]. The main parameters included A3s, G3s, C3s and T3s (frequency of bases corresponding to synonymous codons at the third position), codon adaptation index (CAI), codon deviation index (CBI), optimal codon frequency (FOP), number of effective codons (Nc), number of third codons (G + C) (GC3s), number of genes (G + C) (GC), triple codons, and the correlations of these parameters were analyzed.

### 4.6. Analysis of MdSKP1-Like Member Protein Interactions

Select *Arabidopsis* as the model plant and set the highest confidence to 0.900, and STRING (https://string-db.org/, accessed on 15 March 2023) performed MdSKP1-Like family members protein interaction prediction [51]. It was further embellished with Cytoscape software and set the value to Betweenness (BC) for reconstruction.

### 4.7. Cis-Acting Elements of MdSKP1-Like and Gene Microarray Expression Analysis

The 2kb gene sequence upstream of the *MdSKP1-Like* was submitted to PlantCARE (http://bioinformatics.psb.ugent.be/webtools/plantcare/html/ accessed on 20 March 2023) and relevant gene action element data were obtained. Apple tissue expression data were downloaded from the GEO database for different tissue types [52]. Data sources are flower: M67; fruit: M74; leaf: M14; root: GD; stem: X8877; seedling: GD; and seed: X4442xX2596, so the *SKP1-Like* expression data were extracted according to its gene number. The obtained data were further organized using Excel software, and *SKP1-Like* expression analysis was completed using TBtools.

### 4.8. Quantitative Real-Time Fluorescence PCR (qRT-PCR) Analysis

The CDS sequences of *MdSKP1-Like* were submitted to the website of Bioengineering (Shanghai) Co. for online primer design (Appendix A). cDNA synthesis was performed with the Prime Script RT reagent kit (Perfect Real Time) (TaKaRa). The expression of each gene was quantified using real-time fluorescence quantitative PCR (Mx3005p, Stratagene, La Jolla, CA, USA) using SYBR Green I kit (Takara Biomedical Technology Co., Ltd., Beijing, China) with the GAPDH gene as the internal reference gene. The relative expression of the genes was determined using 2^−ΔΔCT^ [53].

## 5. Conclusions

Collectively, 17 *MdSKP1-like* gene members were identified, predicted to be mainly in the nucleus and cytoplasm. *MdSKP1-Like* gene expression in leaves varied greatly under different hormonal and abiotic stress treatments. qRT-PCR results also showed that *MdSKP1-Like* response to hormones and abiotic stress, under 4 °C treatment for 24 h, the expression of all *MdSKP1-Like* genes was downregulated. The expression of *MdSKP1-Like2* and *MdSKP1-Like10* genes was upregulated after both PEG and NaCl treatments for 24 h, respectively. In addition, the expression of most *MdSKP1-Like* genes was upregulated after 24 h of SA, ABA, and MeJA treatments. We also found that *MdSKP1-Like3* was the most highly expressed gene in all three hormone treatments. These genes can be used as candidate genes for further functional studies of *SKP1-Like* genes involved in abiotic stresses and mediated phytohormone signaling pathways. Overall, these findings provide a comprehensive understanding of the *SKP1-Like* gene family in apples and contribute to the understanding of the role of *SKP1-Like* genes in apple growth, development, and adaptation to abiotic stresses.

## Figures and Tables

**Figure 1 ijms-24-16414-f001:**
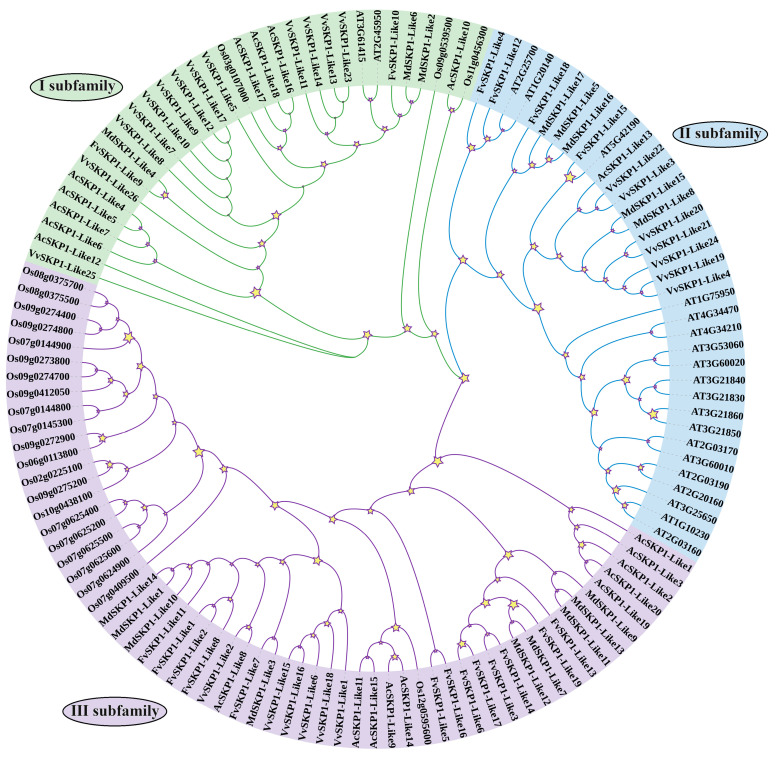
Phylogenetic analysis of *MdSKP1-Like* gene family in apple. Dicots, *Arabidopsis thaliana* (At), *Fragaria vesca* (Fv), *Malus domestica* (Md), *Vitis vinifera* (Vv), and monocots *Ananas comosus* (Ac), *Oryza sativa* (Os) phylogenetic analysis of SKP1-Like proteins. The subfamilies were marked by a colorful background. Stars represent branching nodes of the evolutionary tree.

**Figure 2 ijms-24-16414-f002:**
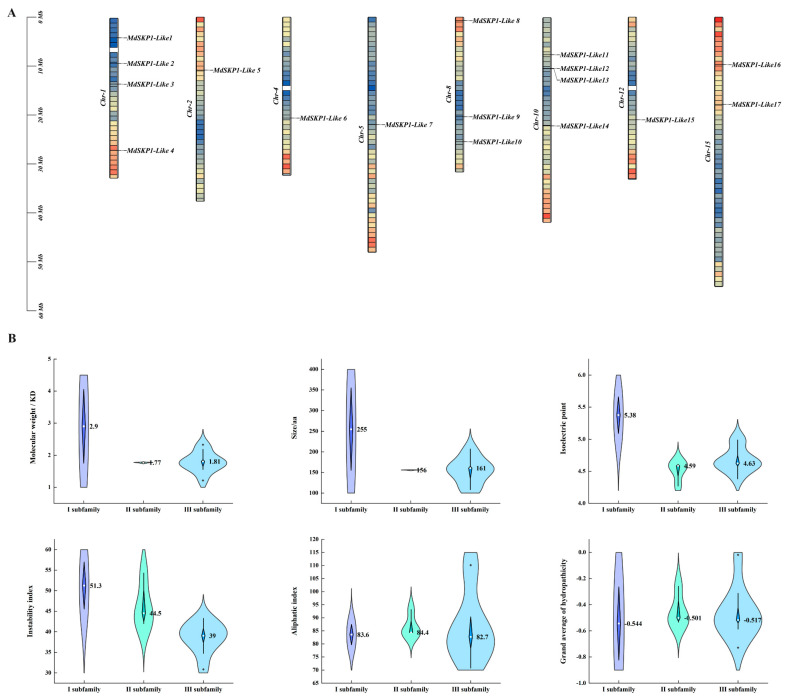
Distribution of *MdSKP1-Like* genes on different apple chromosomes and SKP1-Like protein physicochemical property violin diagram. (**A**) Chromosomes are indicated by colored bars. The position of the *MdSKP1-Like* gene is indicated next to it. The different colors represent gene density, where red indicates high-density regions and blue indicates low-density regions. (**B**) Violin box plots of amino acid size, molecular weight/KD, isoelectric point, instability index, aliphatic index, and grand average of hydropathicity for SKP1-Like proteins. Black dots represent outliers, and numbers with white dots represent averages.

**Figure 3 ijms-24-16414-f003:**
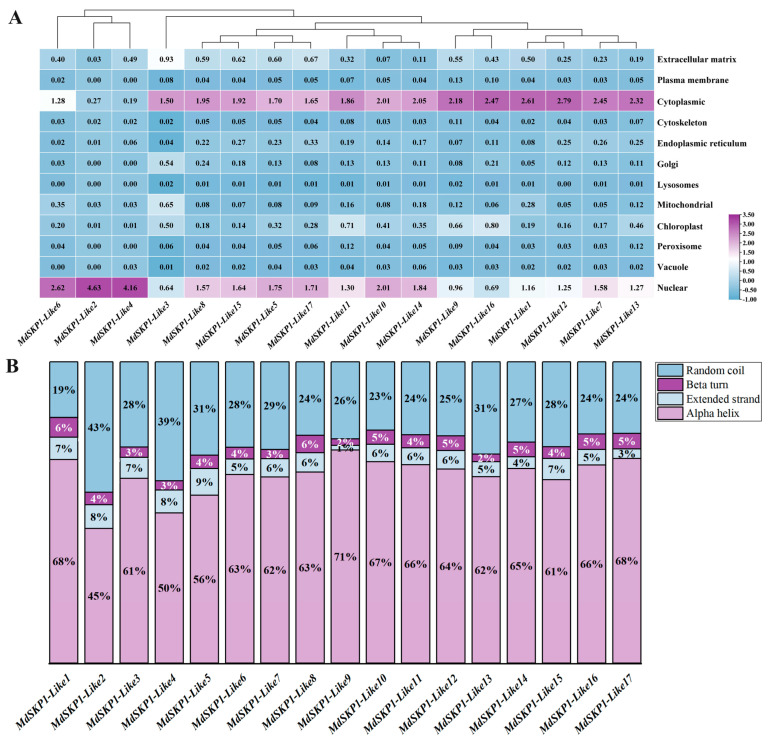
The secondary structure- and subcellular-location prediction of MdSKP1-Like proteins. (**A**) Subcellular location prediction of MdSKP1-Like proteins. Purple for high numbers and blue for low numbers. (**B**) The secondary structure of MdSKP1-Like proteins. Different colors represent different secondary structures.

**Figure 4 ijms-24-16414-f004:**
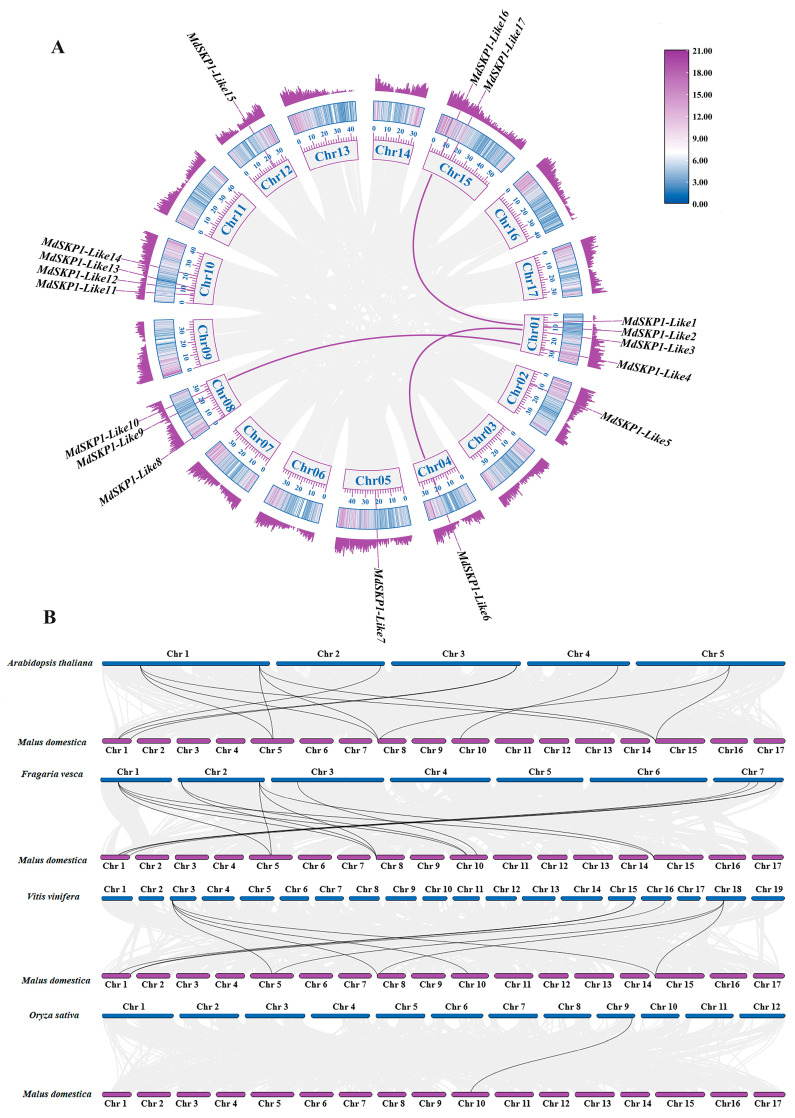
Collinearity relationship analysis of *MdSKP1-Like* gene. (**A**) Collinearity relationship of *SKP1-Like* family genes in apple. In the circles, the *MdSKP1-Like* gene family is labeled on the corresponding chromosome, and the represented collinearity gene pair between the *MdSKP1-Like* genes is indicated by the purple curve. The outermost circle with color and the second outermost circle are expressions of gene density. The purple color has a higher density, and the blue color has the lowest density. (**B**) Collinearity relationships of *MdSKP1-Like* genes between apples and four representative plant species. The gray line in the background indicates apples and their neighboring blocks in the plant genome, while the black line highlights the collinearity *SKP1-Like* gene pairs.

**Figure 5 ijms-24-16414-f005:**
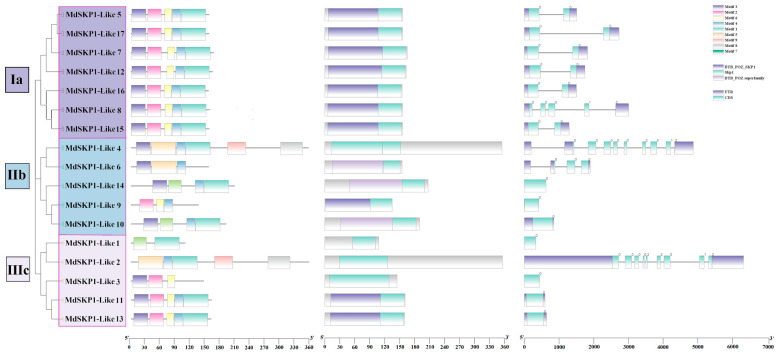
*MdSKP1-Like* gene structure, motif, and structural domain analyses. Subclade Ia is highlighted in purple; subclade IIb is highlighted in blue; and subclade IIIc is highlighted in pink. (**Left**) Conserved motifs; different motifs are marked with different colors; and the numbers above represent different motifs. (**Middle**) Structural domain map; (**Right**) the exon-intron structure of *MdSKP1-Like* gene; and the number represents quantities.

**Figure 6 ijms-24-16414-f006:**
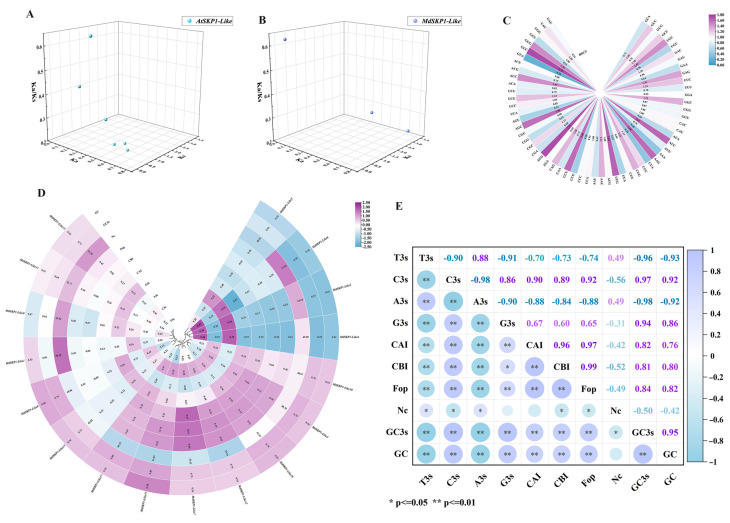
*MdSKP1-Like* gene evolutionary selection pressure and codon usage bias analyses. (**A**) Ka/Ks analysis of *SKP1-Like* collinearity relationship gene pairs. (**B**) Ka/Ks analysis of collinearity relationship gene pairs *SKP1-Like* genes in *Arabidopsis*. (**C**) Relative synonymous codon usage (RSCU). (**D**) Synonymous codon preference and correlation analyses of *MdSKP1-Like* gene. Different colors indicate the magnitude of correlation coefficients; the larger the correlation coefficients are in purple, and the smaller numbers are in blue. U3s, C3s, A3s, G3s, and T3s indicate the U, C, A, and T of the codon third site, respectively. G + C base composition content. CAI, codon adaptation index. CBI, codon preference index. FOP, frequency of optimal codon usage. NC: effective codon number. GC content of the third position of GC3s synonymous codon. GC: GC content. (**E**) Correlation analysis of *MdSKP1-Like* gene codon. Purple indicates a positive correlation; blue indicates a negative correlation; and white indicates no correlation.

**Figure 7 ijms-24-16414-f007:**
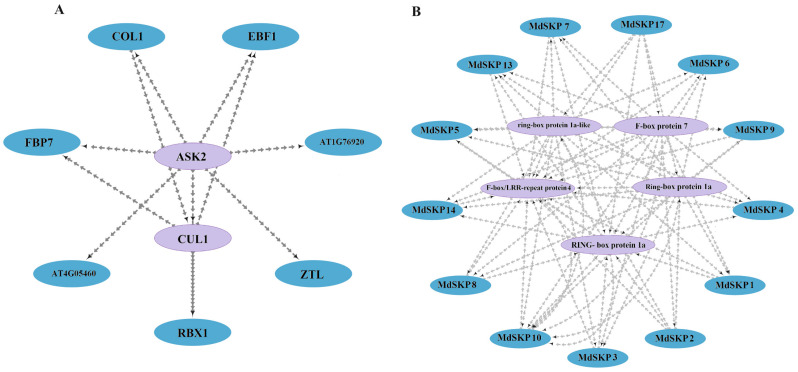
Protein interaction analysis of the SKP1-Like. (**A**) Interaction analysis of ASK2, a homolog of SKP1-Like8, with other proteins. Purple color represents proteins with tighter interprotein interactions, and blue color is the opposite. (**B**) Protein interaction analysis of the SKP1-Like in apple. Blue color represents MdSKP1-like protein members and purple color represents protein members that interact with them.

**Figure 8 ijms-24-16414-f008:**
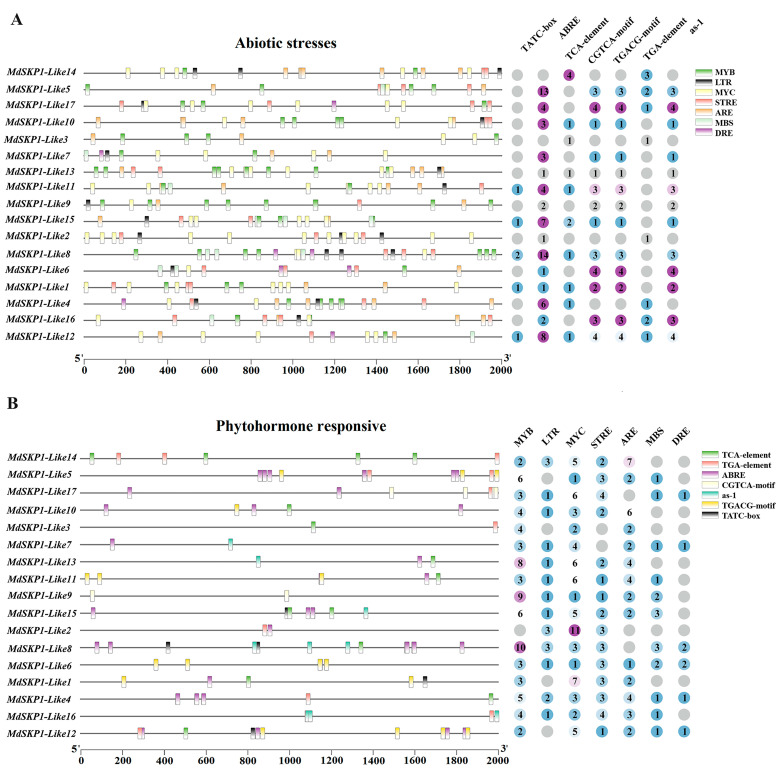
The first 2000 bp of cis-acting elements of 17 *MdSKP1-Like* genes. Different elements are labeled with different colors. (**A**) Abiotic stress-regulating elements predominate. (**B**) Major phytohormone action elements. Numbers represent the number of cis-acting elements.

**Figure 9 ijms-24-16414-f009:**
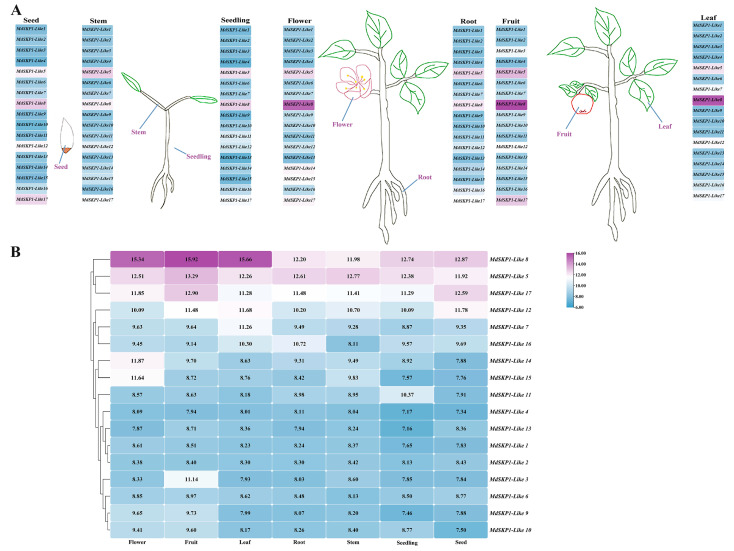
Hierarchical clustering of the expression profiles of the *MdSKP1-Like* gene in seedling and different organs in apple. (**A**) Heat map of the expression of *MdSKP1-Like* genes in seedling and different organs of apple. (**B**) Cluster analysis of heat map data of *MdSKP1-Like* gene based on GEO databases.. Purple and blue represent upregulated or downregulated expression levels, respectively.

**Figure 10 ijms-24-16414-f010:**
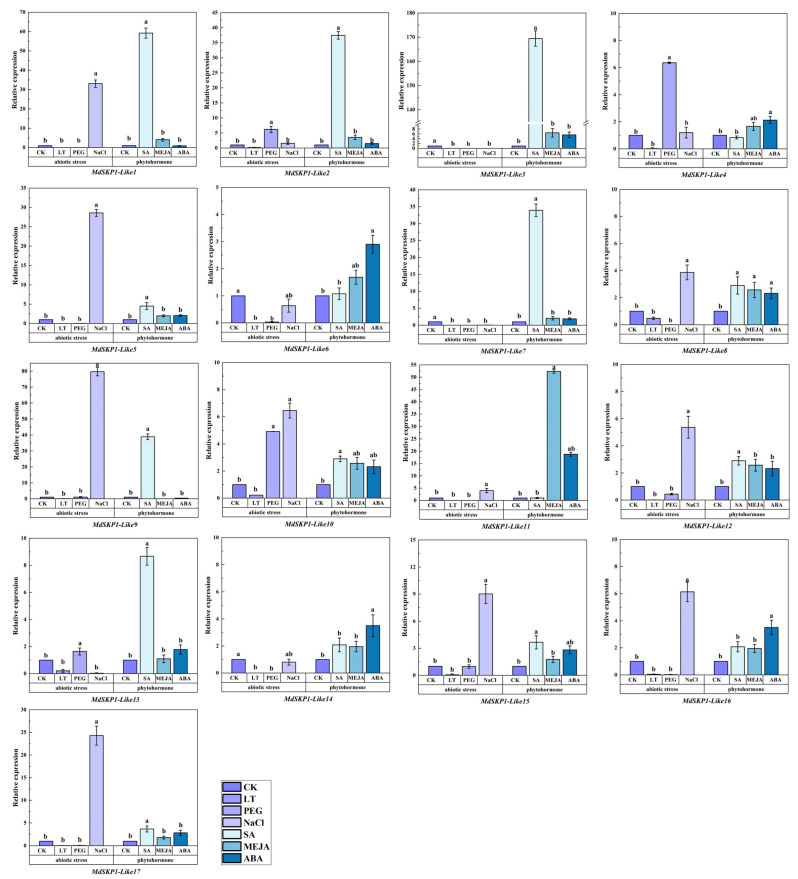
Expression levels of *MdSKP1-Like* gene in abiotic stresses and phytohormones treatments. Expression levels of *MdSKP1-Like* gene in NaCl, 4 °C, PEG, MeJA, ABA, and SA phytohormones treatments. Statistical analysis was performed using one-way ANOVA and Tukey’s honestly significant difference (HSD) test. The expression level of the control group that was not stressed has a value of 1. Black error lines represent the mean ± SE of three biological replicates. Different letters denote significant differences, whereas the same lowercase letters indicate no statistical difference (*p* < 0.05).

## Data Availability

Data will be made available on request.

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
