# Peer review of "Identification and Expression Analysis of the SKP1-Like Gene Family under Phytohormone and Abiotic Stresses in Apple (Malus domestica)"

_ijms, 2023, doi:10.3390/ijms242216414_

Round 1
Reviewer 1 Report
Comments and Suggestions for Authors
Dear Authors,
Authors concentrated on SKP1-Like, a core component of the SCF (Skp1-Cullin-F-box) complex, which is the final step in catalyzing the ubiquitin-mediated protein degradation pathway. Moreover, the SKP1-Like gene family has not been well characterized in response to apple abiotic stresses and hormonal treatments. The 17 MdSKP1-Like gene family members with conserved domain of SKP1 were identified in apples and were distributed on eight chromosomes. The MdSKP1-Like genes located on chromosomes 1, 10, and 15 were highly homologous. Authors stated that, MdSKP1-like genes were divided into 3 subfamilies according to the evolutionary affinities of monocotyledons and dicotyledons. MdSKP1- like members of the same group or subfamily show some similarity in gene structure and conserved motifs.
The introduction part gives the reader sufficient background to analyse Author’s results, but Please, underline the precise aim of the study in presented work;
Authors underline that “Tissue-specific expression analysis showed that most MdSKP1-Like gene family members were highly expressed in flowers and leaves” – this is for sure not “tissue” specific analysis- Please correct the mistake;
It's a pity that the authors only present the predicted subcellular localization – the real effect for selected genes will be more valuable for these results;
Figure 4 is almost unreadable even after enlargement; the same situation refers to figure 8 as well as the upper panel of figure 9- Please, reorganize the figures; Moreover, figure 9 should be separated into two because the levels of genes expression are almost lost;
Tables with primers can be relocated to the supplementary materials;
Materials and methods section is quite good written, but It’s a pity that to these analysis only one reference genes were used, It should be for sure repeated for at least two different references;
The conclusion is largely a repetition of the results – Please, add future prospects coming from obtained results to make these results more visible for the wider audience;
sincerely
Comments on the Quality of English LanguageExtensive editing of English language required.
Author Response
Dear reviewer,Please see the attachment.

Reviewer 2 Report
Comments and Suggestions for Authors
The paper makes a complete study of an important familiy of proteins in a fruit tree. These studies are not so easy to find in the literature and authors have used several bioinformatic tools in depth to get a great understanding of the protein family. In addition they also present a wet experiment, comparing the expression leves under differente conditions of several genes.
I have only found two minor issues.
a) Figure 1. The phylogenetic tree is very complete, but there is a question which remains to be answered. Is there any SKP-1 like member that is only found in malus? do you have evidence of any gene duplication that occurred only in the malus genome? Please comment on this based on data of figures 1 and 6.
b) Figure 10: Is very complete, but the scale in the Y axis has dramatic differences among genes. Does this means that genes whose expression is 100 fold higher that others are going to be more relevant? Is there any information on this? Has this been related to any known QTL, please comment. It would also be interesting to include a supplementary figure in which all the qRT PCR would be depicted with the same sclae in the Y-axis, so it would be easy to visualize which are the most expressed/relevant.
Round 2
Reviewer 1 Report
Comments and Suggestions for Authors
Dear Authors,
Thank you for Authors explanations and adress to future experiments;
I underline once more [Figure 9, panel A and B]- flower, seed, stem, leaf - There are not plant tissues- There are plant organs! - Therefore, this figure and figure captions still needs corrections.
Comments on the Quality of English Language
Minor English corrections
Author Response
Dear reviewer, We sincerely thank the reviewer for your valuable feedback that we have used to improve the quality of our manuscript and thanks for providing us with the opportunity to revise the manuscript again. Please see the attachment.
